# Electrophysiological Properties from Computations at a Single Voltage: Testing Theory with Stochastic Simulations

**DOI:** 10.3390/e23050571

**Published:** 2021-05-06

**Authors:** Michael A. Wilson, Andrew Pohorille

**Affiliations:** 1Exobiology Branch, MS 239-4, NASA Ames Research Center, Moffett Field, CA 94035, USA; Michael.A.Wilson@nasa.gov; 2SETI Institute, 189 Bernardo Ave, Suite 200, Mountain View, CA 94043, USA; 3Department of Pharmaceutical Chemistry, University of California, San Francisco, CA 94132, USA

**Keywords:** computational electrophysiology, electrodiffusion model, stochastic simulations, current–voltage dependence, reversal potential, committor probabilities

## Abstract

We use stochastic simulations to investigate the performance of two recently developed methods for calculating the free energy profiles of ion channels and their electrophysiological properties, such as current–voltage dependence and reversal potential, from molecular dynamics simulations at a single applied voltage. These methods require neither knowledge of the diffusivity nor simulations at multiple voltages, which greatly reduces the computational effort required to probe the electrophysiological properties of ion channels. They can be used to determine the free energy profiles from either forward or backward one-sided properties of ions in the channel, such as ion fluxes, density profiles, committor probabilities, or from their two-sided combination. By generating large sets of stochastic trajectories, which are individually designed to mimic the molecular dynamics crossing statistics of models of channels of trichotoxin, p7 from hepatitis C and a bacterial homolog of the pentameric ligand-gated ion channel, GLIC, we find that the free energy profiles obtained from stochastic simulations corresponding to molecular dynamics simulations of even a modest length are burdened with statistical errors of only 0.3 kcal/mol. Even with many crossing events, applying two-sided formulas substantially reduces statistical errors compared to one-sided formulas. With a properly chosen reference voltage, the current–voltage curves can be reproduced with good accuracy from simulations at a single voltage in a range extending for over 200 mV. If possible, the reference voltages should be chosen not simply to drive a large current in one direction, but to observe crossing events in both directions.

## 1. Introduction

Ion channels are ubiquitous in living systems in which they mediate ion transport across cell walls [1,2,3]. Although all confirmed structures of ion channels are either bundles of α-helices or β-barrels organized around a transmembrane, water-filled pore lined largely with hydrophilic side chains, they markedly differ in their properties. Their activity is regulated by a variety of signals, such as voltage, ligands, pH or mechanical tension. Some channels are made of peptides that barely span the membrane, while others are among the largest protein assemblies in a cell. In terms of ionic conductance, defined as the ratio of ionic current to voltage, channels differ by more than two orders of magnitude and conductance is not correlated with size. For example, the single-channel conductance of a bacterial homolog of pentameric ligand gated ion channels (pLGICs), GLIC, which consists of 317 residues per subunit is 8 pS [4], similar to the lowest conductance level of a channel made of antimicrobial peptide, alamethicin, which is built of 20 amino acids [5]. Another channel-forming peptide trichotoxin (TTX), consisting of 7 helices, each containing 18 residues conducts ions at 850–900 pS [6], which is close to the conductance of mechanosensitve channels MscS containing 250–1100 residues [7], approximately equal to 1 nS [8]. Some channels exhibit exquisite selectivity whereas others are non-selective. Single point mutations can not only markedly affect conductance and selectivity but even render a channel inactive or constitutively open [9,10,11]. How variations in a common, general architecture translate to a variety of electrophysiological behaviors is of great interest not only for understanding regular biological systems but also for explaining a number of diseases associated with improper function of ion channels [12,13,14].

The availability of high-resolution structural models of ion channels has created opportunities to connect structure and function. Molecular dynamics (MD) computer simulations can contribute to this goal by providing mechanistic and thermodynamic descriptions of ion transport that is not readily accessible from experimental studies [15,16,17,18,19,20,21,22,23,24,25,26]. For a recent, comprehensive review, see Flood, et al. [27]. Furthermore, MD simulations can be used to validate experimentally derived structural models, which do not always correspond to the native structures of channels [28], select the native structure among several candidates [29], and predict functional effects of mutations. These simulations, however, have to be validated by demonstrating that they can be used to reproduce measured electrophysiological properties with satisfactory reliability.

Calculating electrophysiological properties from MD simulations with applied voltage can be done simply by way of computing the current across the simulation cell [15,30,31] or counting the number of ions that cross the channel [16,24,25,32]. However, this direct method, especially when applied to obtain I-V curves and reversal potentials, requires significant computational effort, as it involves MD simulations at a number of applied voltages. To obtain the same accuracy, channels with low conductance generally require longer simulations than channels with high conductance. For example, in simulations of TTX, we counted almost 200 K${}^{+}$ crossing events in 900 ns at 50 mV [33], whereas only 23 Na${}^{+}$ crossing events were observed in a 7.7 μs simulation of GLIC at 100 mV (Wilson and Pohorille, unpublished). Since the ionic currents from both simulations appear to obey Poisson statistics, we expect the relative errors in the conductance of K${}^{+}$ in TTX and Na${}^{+}$ in GLIC to be approximately 7% and 20%, respectively. To achieve the same relative errors for currents in GLIC as in TTX, a MD trajectory of over 60 μs in length would be required. This means that calculating the I-V curve might present a considerable challenge. To deal with this challenge, it has been common to improve statistics for ion crossing events by applying high voltages, sometimes substantially above their physiological values, or increasing ionic concentration in bulk solution [15,16,17,34,35,36,37,38]. This approach, however, is fraught with dangers, as it might lead to the disruption of membrane structure or saturation effects for ion entry to a channel [37]. Furthermore, if high voltages are used, I-V curves in physiologically relevant ranges are obtained via interpolation or extrapolation procedures of unknown accuracy [35].

If the motion of ions through the channel can be satisfactorily described as diffusion in the applied electric field and the potential of mean force (PMF) exerted by all other components of the system, which is assumed to be independent of voltage, then the computational effort can be markedly reduced. Several approaches not based on MD take advantage of this description. Methods based on Poisson-Nernst-Planck (PNP) theory rely on solving the electrodiffusion (ED) equation for electrical current in which the mobile ions are represented as a mean-field concentration profile whose distribution and motion is determined by electrostatic forces [39,40,41,42,43,44]. In Brownian Dynamics (BD), channel conductance is calculated by way of solving the Langevin equation in which both short-ranged interactions with a static model of the channel and long-ranged, electrostatic interactions are taken into account [40,45,46,47,48]. In both PNP and BD approaches, electrostatic forces are are obtained from the Poisson equation and the medium is represented as a continuum. From this perspective, we take an approach in which atomistic, dynamic information offered by MD is combined with the efficiency of the ED equation. Instead of carrying out a series of extensive MD calculations of a channel over a range of applied voltages, a substantially reduced set of simulations is combined with the one-dimensional ED model in a steady state. In this approach, the actual electrophysiological properties, such as the current, are calculated from the ED equation, whereas the quantities needed to solve this equation are supplied from MD simulations.

Not all channels conform to the ED model. This model cannot be applied directly if ion crossing events are not statistically independent [49,50], ion diffusion is single file rather than Fickian [51], there are strong binding sites for ions in the channel, the channel changes its structure in response to applied voltage in the range of interest [26] or ions experience saturation in the mouth of the channel. Despite these limitations, it appears that ion movement through many channels satisfies the assumptions of the ED model. A number of small, naturally occurring and synthetic channels and pLGICs belong to this category. The channels discussed in this paper were found to be well-described by the model. More generally, a number of different equations that are special cases of the ED equation, such as Goldman-Hodgkin-Katz (GHK) equation, have been extensively and successfully used as basic tools in experimental and computational electrophysiology for nearly 80 years, for example to determine ionic selectivity from the measured reversal potential [1,52]. Further, basic assumptions of the ED model, such as independence of ion crossing events or Fickian nature of diffusion, can be tested without substantial, additional effort. This was previously done for a number of channels [25].

Previously, we developed an approach to calculate electrophysiological properties from the integrated form of the ED equation [19,24,25]. Instead of MD calculations at several voltages, the system is simulated at a single voltage (or with no applied voltage) to obtain the PMF for each ion in the channel. Subsequently, markedly shorter simulations at voltages of interest are required to determine the densities of the ions near the ends of the channel. Calculating the currents from simulations at these voltages is not required. In addition, ionic diffusivity along the channel has to be determined. Both boundary density values and diffusivity obtained by any of these methods are burdened with errors, contributing to inaccuracies in the calculated currents.

Recently, we developed two formalisms for calculations of electrophysiological properties, including I-V curves and reversal potential, from a single MD simulation at one voltage [33]. From this simulation, the PMF, nonequilibrium density profiles and committor probabilities for ions in the channel are obtained and used to calculate currents at different voltages after appropriate transformations of the ED equation. Additional calculations to obtain the density boundary terms at different voltages and diffusivity are no longer needed. These formalisms were tested on a simple model of the TTX channel, comprised of 7 straight α-helices, each containing 18 amino acids [6], and were shown to perform very well. The improved efficiency of this novel approach derives from the fact that only one MD simulation instead of multiple ones is needed to obtain the I-V curve or reversal potential.

As is the case for any new approach, it is essential to establish the intrinsic accuracy of our formalism. This is the goal of this paper. Specifically, we focus on the question: how reliable are our approaches to calculating electrophysiological properties, independent of other sources of errors, such as inaccuracies in force fields and insufficient simulation times? Separating errors due to the proposed methods from other error is not simple. It cannot be done through a direct comparison with experiments because, for example, of inaccuracies due to force fields. In principle, it can be done via comparison with accurate MD simulations of the PMF and currents at several applied voltages, but, in practice, it is expensive to obtain sufficiently accurate free energies and currents. Although PMFs for ions in a number of channels were obtained from MD simulations, in most cases no estimates of errors were provided [21,23,26,53,54,55,56,57,58,59,60,61]. In a few cases in which errors are available by way of either direct estimates or comparisons of PMF obtained via different methods [20,24,25,62,63,64,65,66,67,68] they are of the order of 0.2–0.7 kcal/mol, which is similar to what is expected to be the intrinsic accuracy of the formalisms studied here. This means that if there were differences between electrophysiological properties obtained from MD simulations at several applied voltages and reconstructed from simulations at a single voltage it would not be possible to determine whether these differences were due to insufficient accuracy of the simulations or to inaccurate reconstruction from the new methods. For these reasons, we take a different approach.

We assume that the ED model describes ion transport with satisfactory accuracy and that the underlying PMF is known. Then, the ED equation is solved many times by way of stochastic simulations to ascertain how statistical errors depend on the number of stochastic trajectories. Even though the stochastic simulations employed here do not involve time explicitly, the number of trajectories considered in a stochastic dataset can be related to the MD simulation time. In a simple case of TTX, we demonstrate that this can be done consistently. For each set of stochastic trajectories, the PMF and the electrophysiological properties at different applied voltages are reconstructed by way of the new theoretical approaches considered here. Due to the stochastic nature of each solution to the ED equation and the limited number of trajectories in each set, which can be related to a specific simulation time, the quantities of interest obtained from each reconstruction differ between themselves and from the accurate values associated with the underlying PMF. Further, the calculated quantities also depend on the theoretical formalism used. If a sufficient number of trajectories has been generated, statistical errors on the quantities of interest can be estimated as a function of the number of trajectories or equivalently, simulation time and performance of each theoretical approach can be systematically assessed. For sets with a large number of trajectories, which corresponds to long simulation times in MD, the underlying PMFs will be reconstructed accurately. For sets with a smaller number of trajectories, the accuracy will not be as good and is expected to deteriorate as the number of trajectories is reduced. A similar systematic study cannot easily be done in practice by way of MD simulations because the computational effort to generate many MD trajectories of different length would have been prohibitive. Furthermore, no analytical method for error analysis exists for this problem.

In principle, this type of analysis can be carried out for any underlying PMF, even if it is unrelated to real ion channels. This is, however, not the direction that we pursue. Instead, we use the PMFs that we previously obtained from simulations of three actual channel models and, for the purpose of this study, assume that they are accurate. The models were selected such that they differ in size, pore structure, conductance and selectivity. The first model is TTX, which exhibits relatively high conductance, very little structure inside the pore and a weak selectivity for cations.

The second model is the high-resolution NMR structure proposed by OuYang et al. [69] for a hexameric channel p7 from the hepatitis C virus. Each subunit consists of 63 amino acids. The model has an unusual architecture not found in any other channel. The channel does not exist as a bundle of α-helices, which is the most common structural motif among membrane proteins, but instead forms an interlocked structure in which each subunit assumes a horseshoe conformation with each side comprised of a short, α-helical section. Because of these atypical features there have been concerns about the veracity of this model [28]. Recently, we calculated conductance and ionic selectivity of this model by way of MD simulations and showed that both properties differ significantly from those measured experimentally (Shannon et al., unpublished). Specifically, in contrast to the electrophysiological data, the model exhibits high conductance and strong selectivity for Cl${}^{-}$ over K${}^{+}$. These results strongly suggest that the proposed model does not represent the native structure of the channel, demonstrating that computational electrophysiology can be used not only to support but also to disprove structural models of ion channels.

The third model is based on the crystal structure of a pentameric, cation-selective ion channel, GLIC, from a cyanobacterium Gloeobacter [70]. This channel is a bacterial homolog of receptors belonging to the family of pentameric ligand-gated ion channels. Its main electrophysiological characteristics are low-conductance (9.3 pS) and strong selectivity for cations [70]. Molecular models of all three channels are shown in the Appendix A.

Both the p7 and GLIC models have a markedly more varied pore structure than TTX and, consequently, a more complex PMF. Although we will use the names of these three channels further in the text, we do not claim that the underlying PMFs faithfully represent the PMFs for these channels in their native open forms (this does not appear to be the case for p7) and, therefore, we do not compare the electrophysiological properties calculated from stochastic simulations with the same properties obtained experimentally. Instead, we fully concentrate on assessing the accuracy of the underlying theory.

## 2. Theory and Method

In this section, we briefly outline the theory behind three different approaches to calculating the PMF and electrophysiological characteristics of an ion channel. Two of them require simulations at only one applied voltage. A more detailed derivation of the basic formulas, which follows closely the earlier development [33], is provided in Appendix A. Next, we describe how the properties of interest can be obtained from stochastic simulations under the assumptions of the ED model specified in the introduction. Note that while the theory is developed in the context of MD simulations, here, we use the results of the theory to compute the PMF and I-V curves from density profiles and committor probabilities that were obtained from stochastic simulations.

### 2.1. Calculating the Potential of Mean Force

If the concentrations of ions on both sides of the membrane and the applied voltage remain constant in time, the system is in a steady state, which means that the flux of ions through the channel, *J*, is also constant in time. These are the conditions most often considered in both experiments and simulations aimed at extracting electrophysiological properties of channels. Then, the one-dimensional ED equation for a given type of ions can be written as
(1)$$J=-D\left(z\right)\left(\frac{d\rho (z)}{dz}+\beta \rho \left(z\right)\frac{dE(z)}{dz}\right),$$
where D(z) is the diffusivity that, in general, depends on position *z* along the reaction coordinate z. For a transmembrane channel embedded in the membrane located in the x,y-plane, a convenient reaction coordinate is the position of an ion along the pore of the channel, which can be measured along the z-coordinate. ρ(z) is the line density of ions, which is usually recorded as a histogram in computer simulations. $\beta =k_{B}T$, where $k_{B}$ is the Boltzmann constant and *T* is temperature. E(z) is given by
(2)E(z)=A(z)+qV(z),
where A(z) is the PMF, V(z) is the applied voltage and *q* is ionic charge. In a constant electric field, $E^{el}$, acting along z, which is the most frequent experimental condition,
(3)$$V\left(z\right)=E^{el}\left(z-z_{a}\right).$$
Even though the electric field is applied across the whole system [15,30], it acts only between $z_{a}$ and $z_{b}$ in the non-polar phase, which has been identified as corresponding to the hydrophobic core of the membrane [25,33]. Thus, electric field is a boxcar function that is equal to $E^{el}$ in the range [$z_{a},z_{b}$] and zero otherwise. This can be formally written as $E^{el}\left[H\left(z-z_{a}\right)-H\left(z-z_{b}\right)\right]$, where *H* is the Heaviside function. Although we will not use this notation for simplicity, the range in which $E^{el}$ is non-zero has to be kept in mind.

Integrated with the integrating factor $\exp\left[\beta E(z)\right]$ and resolved for *J*, the ED equation takes the form
(4)$$J=\frac{\rho \left(z_{min}\right)\exp\left[\beta E(z_{min})\right]-\rho \left(z_{max}\right)\exp\left[\beta E(z_{max})\right]}{\int_{z_{min}}^{z_{max}}\frac{\exp\left[\beta E(z)\right]}{D(z)}dz}.$$
For a system in a steady state, *J* does not formally depend on the limits of integration $z_{min}$ and $z_{max}$. This means that these limits do not have to coincide with the edges of the channel. In practice, the limited precision of MD simulations introduces some dependence on the limits of integration, as analyzed elsewhere [25].

To calculate *J* from this equation, E(z) has to be known, which in turn requires determining A(z). This can be done in equilibrium simulations in the absence of voltage. A host of methods exist for this purpose [71,72,73]. A(z) can be also calculated from non-equilibrium simulations at an applied voltage. If the ED equation is integrated with the integrating factor 1/ρ(z) then
(5)$$J=\frac{\ln\frac{\rho (z_{min})}{\rho (z_{max})}-\beta \left[A\left(z_{max}\right)-A\left(z_{min}\right)+qE^{el}\left(z_{max}-z_{min}\right)\right]}{\int_{z_{min}}^{z_{max}}\frac{1}{D(z)\rho (z)}dz}.$$
Since *J* is independent of the limits of integration, $z_{max}$ can be substituted by *z*. After simple rearrangements, it yields a formula for the PMF relative to its value at $z_{min}$, $\Delta A\left(z,z_{min}\right)=A\left(z\right)-A\left(z_{min}\right)$
(6)$$\Delta A\left(z,z_{min}\right)=-k_{B}T\left[\ln\frac{\rho (z)}{\rho (z_{min})}+J\int_{z_{min}}^{z}\frac{1}{D\left(z^{\prime }\right)\rho \left(z^{\prime }\right)}dz^{\prime }\right]-qE^{el}\left(z-z_{min}\right).$$
We call this method for determining PMF the Integrated Electrodiffusion Equation Method (IEEM).

To solve Equations (Equation 5) and (Equation 6), D(z) has to be known in the full range of *z*. D(z) can be determined by way of calculating the mean square displacement of the ion at several points along the channel obtained from a series of short MD trajectories after subtracting the PMF [19], from the force-force autocorrelation function acting on a stationary ion at different positions in the channel [74], or by way of a Bayesian fitting method [75,76,77]. See Appendix A for a discussion of how diffusivity was computed in our MD simulations.

Once $\Delta A(z_{max},z_{min})$ and D(z), which are both assumed to be independent on voltage, are known, the boundary density terms $\rho (z_{min})$ and $\rho (z_{max})$ have to be obtained from either MD or stochastic simulations at each voltage of interest. Since the full knowledge of ρ(z) is not needed, these simulations can be markedly shorter than simulations to determine the PMF. Then, $\Delta A(z_{max},z_{min})$, D(z), $\rho (z_{min})$ and $\rho (z_{max})$ are used in Equation (Equation 5) to calculate *J* at a given voltage. Previously, we demonstrated that this method performs satisfactorily for simple channels [19,24,25].

Recently, we developed two alternative approaches to calculating the PMF and electrophysiological properties that require markedly less computational effort [33]. Both rely on separating the total ionic current, *J*, to currents moving in two opposite directions – from $z_{min}$ to $z_{max}$ and from $z_{max}$ to $z_{min}$. We abbreviate them $J^{f}$ and $J^{b}$ and call them forward and backward currents, respectively.
(7)$$J^{f}=\frac{\rho^{f}\left(z_{min}\right)\exp\left[\beta E(z_{min})\right]-\rho^{f}\left(z\right)\exp\left[\beta E(z)\right]}{\int_{z_{min}}^{z}\frac{\exp\left[\beta E(z^{\prime })\right]}{D(z^{\prime })}dz^{\prime }},$$
(8)$$J^{b}=\frac{\rho^{b}\left(z\right)\exp\left[\beta E(z)\right]-\rho^{b}\left(z_{min}\right)\exp\left[\beta E(z_{mim})\right]}{\int_{z_{min}}^{z}\frac{\exp\left[\beta E(z^{\prime })\right]}{D(z^{\prime })}dz^{\prime }},$$
(9)$$J=J^{f}-J^{b}.$$
Here, $\rho^{f}\left(z\right)$ and $\rho^{b}\left(z\right)$ are densities of ions that entered the range $\left[z_{min},z_{max}\right]$ at $z_{min}$ and $z_{max}$, respectively. We assume that both forward and backward currents are in a steady state and, therefore, their values do not depend on the limits of integration. This allows for setting the upper limit to $z_{min}<z\leq z_{max}$.

Assume that $J^{f}>0$ and $J^{b}>0$ and take the ratio of Equation (Equation 7) to Equation (Equation 8). This yields
(10)$$\frac{J^{f}}{J^{b}}=\frac{\rho^{f}\left(z_{min}\right)-\rho^{f}\left(z\right)\exp\left[\beta \Delta E(z,z_{min})\right]}{\rho^{b}\left(z\right)\exp\left[\beta \Delta E(z,z_{min})\right]-\rho^{b}\left(z_{min}\right)},$$
where
(11)$$\Delta E\left(z,z_{min}\right)=E\left(z\right)-E\left(z_{min}\right).$$
Combined with Equations (Equation 2) and (Equation 3), Equation (Equation 10) can be solved for $\Delta A(z,z_{min})$
(12)$$\Delta A\left(z,z_{min}\right)=-k_{B}T\ln\frac{J^{b}\rho^{f}\left(z\right)+J^{f}\rho^{b}\left(z\right)}{J^{b}\rho^{f}\left(z_{min}\right)+J^{f}\rho^{b}\left(z_{min}\right)}-qE^{el}\left(z-z_{min}\right).$$
From this equation it follows that the PMF can be obtained from non-equilibrium simulations at applied electric field $E^{el}$ simply from an average of ion densities in the forward and backward directions weighed by the backward and forward currents, respectively. We call this method for calculating the PMF the Current-Weighted Density Method (CWDM). Knowledge of diffusivity is not necessary in CWDM. The denominator in the argument of the logarithmic function sets the reference value of the PMF at $z_{min}$.

If we abbreviate the number of crossing events in forward and backward direction as $n^{f}$ and $n^{b}$, respectively, then, assuming that crossing events are governed by the Poisson statistics, the corresponding errors will be approximately $1/\sqrt{(}n^{f})$ and $1/\sqrt{(}n^{b})$. This means that if $n^{f}$ or $n^{b}$ is small, $\Delta A(z,z_{min})$ calculated from Equation (Equation 12) may become inaccurate. Thus, we developed another, related theoretical approach for determining the PMF from non-equilibrium simulations that does not suffer from this disadvantage. Since it requires calculating committor probability, P(z), we will call it the Committor Probability Method (CPM). For a diffusive process considered here, P(z) referenced to the forward direction is defined as the probability that a particle (ion) in position *z* will reach $z_{max}$ before it reaches $z_{min}$. P(z) can be calculated either directly during computer simulations or in post-processing, as described in Appendix A. A general discussion of committor probabilities in more than one dimension and their application to chemical kinetics can be found elsewhere [78,79,80,81].

The PMF can be calculated from ion densities in the forward or the backward direction. The corresponding formulas are
(13)$$\exp\left[\beta \Delta E\left(z,z_{min}\right)\right]=\frac{\rho^{f}\left(z_{min}\right)\left[1-P(z)\right]}{\rho^{f}\left(z\right)-\rho^{f}\left(z_{max}\right)},$$
(14)$$\exp\left[\beta \Delta E\left(z,z_{min}\right)\right]=\exp\left[\beta \Delta E\left(z_{max},z_{min}\right)\right]\frac{\rho^{b}\left(z_{max}\right)P\left(z\right)}{\rho^{b}\left(z\right)-\rho^{b}\left(z_{min}\right)}.$$

Their derivation closely follows our earlier work [33] and is given in Appendix A.

Both equations allow for calculating the same quantity— the PMF. Individually, each of them is not expected to be accurate in the full $\left[z_{min},z_{max}\right]$ range of *z*, especially away from the entry point. Specifically, as *z* becomes close to $z_{max}$ both $\rho^{f}\left(z\right)-\rho^{f}\left(z_{max}\right)$ and 1−P(z) approach zero. Since numerical inaccuracies in Equations (Equation 13) and (Equation 14) affect mainly the opposite sides of the $\left[z_{min},z_{max}\right]$ range, these two equations can be profitably combined. Then, $\rho^{f}\left(z\right)-\rho^{f}\left(z_{max}\right)$ and $\rho^{b}\left(z\right)-\rho^{b}\left(z_{min}\right)$ can be considered as two biased distributions representing the same unbiased distribution h(z). The problem of merging them to reconstruct h(z) such that statistical error on $\Delta A(z,z_{min})$ is minimized can be solved by way of the Weighted Histogram Analysis Method (WHAM) [82]. This yields the following formula for reconstructing the PMF from non-equilibrium simulations:(15)$$\Delta A\left(z,z_{min}\right)=C-k_{B}T\ln\left[\frac{h(z)}{\rho^{f}\left(z_{min}\right)\left(1-P\left(z\right)\right)P\left(z\right)}\right]-qE^{el}\left(z-z_{min}\right),$$
where neither *C*, which is a constant that only shifts the energy scale, nor $\rho^{f}\left(z_{min}\right)$, which is independent of *z* and is needed to ensure that the PMF at $z=z_{min}$ is equal to zero, influences the shape of $\Delta A(z,z_{min})$. Similarly to Equation (Equation 12), no knowledge of diffusivity is required.

Typically, MD simulations would be carried out on the channel system of interest at some applied voltage *V*. From this, the committor probability, P(z) and the 1-sided density profiles, $\rho^{f}\left(z\right)$ and $\rho^{b}\left(z\right)$, and the number of forward and backward crossing events would be determined, and used to calculate the forward and backward fluxes, $J^{f}$ and $J^{b}$, respectively. The PMF can be determined from either CWDM (Equation (Equation 12)) or CPM (Equation (Equation 15)). As will be discussed later, we created synthetic data sets of $10^{6}$–$10^{8}$ stochastic trajectories. For each data set, we calculate the same quantities that would be calculated in MD, P(z), $\rho^{f}\left(z\right)$ and $\rho^{b}\left(z\right)$, and then use these to calculate the PMF. Note that the free energy depends on ratios of density profiles, so the absolute normalization of the density profiles is not important. Similarly, the CWDM requires only ratios of the forward and backward currents, so the magnitudes are not required.

### 2.2. Calculating I-V Dependence from Simulation at a Single Voltage

If the PMF, the current, $J_{\mu }$, and the density, $\rho_{\mu }\left(z\right)$, or the committor probability, $P_{\mu }\left(z\right)$, are known from simulations at an applied voltage, $\Delta V_{\mu }$, the current, $J_{\nu }$, at a different voltage $\Delta V_{\nu }$ can be obtained without any calculations at this voltage. This allows for reconstructing the I-V curve from simulations at a single voltage.

If Equation (Equation 1) is integrated with the same integrating factor, $\exp\left[\beta E_{\nu }\left(z\right)\right]$, for both voltages, $\Delta V_{\mu }$ and $\Delta V_{\nu }$, we obtain
(16)$$\begin{array}{cc} \; & J_{\mu }=\\ \; & \frac{\rho_{\mu }\left(z_{min}\right)\exp\left[\beta E_{\nu }\left(z_{min}\right)\right]-\rho_{\mu }\left(z_{max}\right)\exp\left[\beta E_{\nu }\left(z_{max}\right)\right]+\beta q\left(E_{\nu }^{el}-E_{\mu }^{el}\right)\int_{z_{min}}^{z_{max}}\rho_{\mu }\left(z\right)\exp\left[\beta E_{\nu }\left(z\right)\right]dz}{\int_{z_{min}}^{z_{max}}\frac{\exp\left[\beta E_{\nu }\left(z\right)\right]}{D(z)}dz}, \end{array}$$
and
(17)$$J_{\nu }=\frac{\rho_{\nu }\left(z_{min}\right)\exp\left[\beta E_{\nu }\left(z_{min}\right)\right]-\rho_{\nu }\left(z_{max}\right)\exp\left[\beta E_{\nu }\left(z_{max}\right)\right]}{\int_{z_{min}}^{z_{max}}\frac{\exp\left[\beta E_{\nu }\left(z\right)\right]}{D(z)}dz}$$
The latter but not the former equation is the standard integrated form of the ED equation, Equation (Equation 4).

If we take the ratio of currents $J_{\mu }/J_{\nu }$ then, after some algebra given in Appendix A we obtain
(18)$$\frac{J_{\mu }}{J_{\nu }}=1+\beta q\left(E_{\nu }^{el}-E_{\mu }^{el}\right)\int_{z_{min}}^{z_{max}}f_{\mu }\left(z\right)\exp\left\{\beta q\left[V_{\nu }\left(z\right)-V_{\mu }\left(z\right)\right]\right\}dz,$$
where
(19)$$f_{\mu }\left(z\right)=\exp\left[\beta \Delta E_{\mu }\left(z,z_{min}\right)\right]\frac{\rho_{\mu }\left(z\right)}{\rho_{\mu }\left(z_{min}\right)}.$$
or
(20)$$f_{\mu }\left(z\right)=1+\exp\left[\beta \Delta E_{\mu }\left(z,z_{min}\right)\right]\frac{\rho_{\mu }\left(z_{max}\right)}{\rho_{\mu }\left(z_{min}\right)}-P_{\mu }\left(z\right),$$
depending on whether it is preferred to calculate $J_{\nu }$ from ion density, $\rho_{\mu }\left(z\right)$, or committor probability, $P_{\mu }\left(z\right)$. In both instances, neither diffusivity nor quantities at the applied voltage $\Delta V_{\nu }$ are needed. Equation (Equation 19) is expected to be less accurate that Equation (Equation 20) because $\Delta E_{\mu }\left(z,z_{min}\right)$ and $P_{\mu }\left(z\right)$ that enter the latter equation are estimated on the basis of both forward and backward simulations, whereas $\rho_{\mu }\left(z\right)$ in the former equation is a one-sided density that looses accuracy away from the entry point.

Unlike the free energy, Equation (Equation 18) gives only ratios of forward or backward currents with respect to a reference voltage. Consequently, to calculate the I-V curves, we need currents at this reference voltage.

### 2.3. Stochastic Simulations

The electrodiffusion equation was solved by generating trajectories on a free energy surface E(z) that included the PMF and applied electric field with diffusivity D(z) or average diffusion coefficient, <D>, at temperature *T* [83,84]. This allowed us to generate the channel crossing statistics, density profiles and committor probabilities for the ions for this free energy surface. As the crossing events that we have observed in MD simulations appear to obey Poisson statistics, independently for both ions, we consider the ED equation for each ion separately. Then, we calculated statistical errors in recovering the underlying PMF and the I-V curves as functions of the number of trajectories.

As above, we define the channel boundaries as $z_{min}$ and $z_{max}$, and absorbing boundary conditions were located at these points. Trajectories were initiated at a point just inside the boundary at either $z_{min}$ for forward trajectories or $z_{max}$ for backward trajectories, and propagated until they reached either of the absorbing boundaries. Forward and backward trajectories are considered separately as we are interested in the 1-sided density profiles and committor probabilities, as well as their 2-sided combination. A trajectory that crossed from $z_{min}$ to $z_{max}$ is said to be a crossing trajectory in the forward direction. Similarly, trajectories that cross from $z_{max}$ to $z_{min}$ are crossing trajectories in the backward direction. For simplicity, these will be referred to as forward or backward crossing events. Since the trajectories are initiated near the absorbing boundaries, the majority of trajectories in either direction do not cross, but they do contribute to the 1-sided density profiles and first-passage statistics that are used to compute the committor probabilities (see Appendix A for further details).

The number of trajectories initiated per data set were typically $10^{6}$, $10^{7}$ or $10^{8}$, further abbreviated as N6, N7 and N8, respectively. These numbers were chosen because the average number of crossing events for PMFs corresponding to the models of TTX and p7 observed for $N=10^{6}$ is of the same order of magnitude as the numbers of crossing events observed in our MD simulations of 0.5–2 μs. For the cation-selective GLIC channel, in which the free energy barrier to permeation of Na${}^{+}$ is markedly higher, simulations with $10^{6}$ trajectories yielded too few crossing events to be useful. In this case, the number of crossing events observed at the N7 level approximately corresponds to the number seen in MD simulations of 8 μs. See Appendix A for details of the MD simulations.

The free energy surfaces for the stochastic simulations were obtained by adding the voltage ramp to the PMFs. We use a set of PMFs from our MD calculations. For our problem, the PMFs are the equilibrium free energy surfaces for moving an ion along the 1-dimensional reaction coordinate of the ion with respect to the center-of-mass of the protein channel, at the bulk ion densities of the MD. For this study, we used average diffusion coefficients, <D> obtained by averaging the diffusivities estimated from MD (Appendix A). Strict matching of diffusivity is not necessary for the primary purpose of this study, but it provides a more realistic connection between statistical errors estimated in stochastic simulations and time scales of MD simulations. Additional details are given in Appendix A. The forward and backward ion density profiles were obtained from histograms of either the forward or backward trajectories in each data set. Committor probabilities (Appendix A) were calculated from the first passage statistics of the forward and backward trajectories. The density histograms and committor probabilities were computed for each data set, and not as an average over the individual trajectories in the data set. Averages were constructed over multiple data sets.

## 3. Results and Discussion

### 3.1. Connection with Molecular Dynamics

To compute the currents for the I-V curves from stochastic simulations, some connection to MD is required. MD simulations provide both forward and backward ion trajectories as part of the simulation, unless the channel is strongly rectifying or a large voltage is applied. The net current due to a particular ion is $J=J^{f}-J^{b}$ (Equation (Equation 9)). The total current is the sum of these net currents over all types of ions. As mentioned in the introduction, *J* can be obtained from MD simulations by way of combining the fluxes from forward and backward crossing events or calculating the ionic displacement currents. In stochastic simulations, only the former method can be used. Therefore, we tested whether both method yield the same results for MD and found that this was indeed the case. Both methods and the results of the tests are described in Appendix A.

In addition, the detailed balance condition connecting forward and backward crossing events has to be satisfied. In MD simulations this problem is implicitly solved: if there is no external voltage, simulations of transmembrane systems will exhibit no net current to within statistical errors, which means that the number of forward and backward crossing events is equal, again to within statistical errors. In stochastic simulations, detailed balance also has to be satisfied, which means that trajectories in both directions have to be combined with the correct weights. To determine these weights, we carried out sets of $10^{8}$ simulations with no applied voltage to obtain the well converged, average numbers of forward and backward crossing events. From these simulations, the ratio of forward to backward trajectories that satisfies the detailed balance condition was established and subsequently used to compute the density profiles, committor probabilities and the PMFs at different voltages.

Once the ratio of forward to backward trajectories needed to satisfy the detailed balance condition is known, the average numbers of crossing events in both directions, $n^{f}\left(\Delta V\right)$ and $n^{b}\left(\Delta V\right)$, can be obtained from stochastic simulations at a given voltage ΔV. This, however, is still insufficient to determine currents; additional information about time scales is required. This can be obtained from a MD simulation of the system. We abbreviate the number of forward and backward crossing events observed in MD simulations at applied voltage, $\Delta V_{ref}$, as $m^{f}$ and $m^{b}$. Then, the length of the MD trajectory, $t_{MD}$, can be used to estimate a stochastic time, $t_{S}$, corresponding to the number of stochastic trajectories that produced $n^{f}\left(\Delta V_{ref}\right)$ and $n^{b}\left(\Delta V_{ref}\right)$ crossing events at the voltage $\Delta V_{ref}$. A simple way to make such estimate is to use the number of crossing events in one direction. It is, of course, recommended to choose the direction that provides better statistics. Assuming that there are more forward than backward crossing events in the MD simulations,
$$t_{S}=t_{MD}\frac{n^{f}\left(\Delta V_{ref}\right)}{m^{f}}.$$
If the backward events dominate, $t_{S}$ would be estimated using $n^{b}\left(\Delta V_{ref}\right)$ and $m^{b}$. Once $t_{S}$ has been determined, the stochastic currents at voltage ΔV can be calculated:$$J_{S}\left(\Delta V\right)=J_{S}^{f}\left(\Delta V\right)-J_{S}^{b}\left(\Delta V\right)=\left[n^{f}\left(\Delta V\right)-n^{b}\left(\Delta V\right)\right]/t_{S}.$$
where $J_{S}\left(\Delta V\right)$, $J_{S}^{f}\left(\Delta V\right)$ and $J_{S}^{b}\left(\Delta V\right)$ are the total, forward and backward currents at voltage ΔV.

### 3.2. Committor Probabilities

The committor probabilities for p7 are shown in Figure 1. The committor probabilities in Figure 1a have been calculated from Appendix A, in which ions arriving at *z* from both sides are included. The statistical errors associated with P(z) at different voltages are small, even at the N6 level. As can be seen in Figure 1b, this is not the case for one-sided P(z), obtained using Appendix A. This is due to the decreasing number of ions from one direction as they approach the opposite side of the channel. The inset of Figure 1b shows the numbers of first-passage events from the forward and backward calculations as well as the combined number of events. At 140 mV, the statistics are satisfactory only in the forward direction, in which most of crossing events occur, whereas no reliable probabilities are obtained for the backward direction over the full range of *z*. The opposite is true for −140 mV; P(z) in the forward direction is unreliable. Thus, combining information about P(z) in both directions is preferable whenever possible.

As voltage changes from −140 mV to 140 mV, the position of the transition state for K${}^{+}$ permeation through p7, defined as the x,y-plane at which P(z)=0.5, Ref. [85] shifts substantially and systematically from 7 Å to −13 Å with respect to the center of mass of the membrane. Such large shifts, however, are not universal. As we have shown in the example of TTX [33], the position of the transition state changes markedly less with voltage if the underlying PMF is strongly peaked.

Calculating P(z) for GLIC is more difficult. This is a slow channel and even at the N7 level, which approximately corresponds to a MD trajectory of 8 μs in length (see Appendix A, the number of crossing events is small. At 100 mV only an average number of 0.5 forward and 29 backward crossing events were observed. In particular, N7 simulations of forward trajectories frequently produce no crossing events. At the same voltage, P(z) in the backward direction is often equal to 1 over a relatively wide range of several Å near $z_{max}$, which means that all ions that reached this range exit the channel at $z_{max}$. In such circumstances, calculation of P(z) from Appendix A is no longer possible. A different approach is needed.

Direct calculation of the committor probability requires that some number of trajectories successfully cross the channel, $N^{b}\left(z_{min}\right)$> 0 and $N^{f}\left(z_{max}\right)$> 0. If one of these conditions is not met, for example, if $N^{f}\left(z_{max}\right)$ = 0, then $P^{f}\left(z\right)=N^{f}\left(z_{max}\right)/N^{f}\left(z\right)$ = 0. If we consider position $z^{\prime }$ ($z^{\prime }$ < $z_{max}$) at which $N^{f}\left(z^{\prime }\right)$> 0, then we can write the committor probability P(z) = $\alpha N^{f}\left(z^{\prime }\right)/N^{f}\left(z\right)$, where α is unknown, though formally would be equal to $N^{f}\left(z_{max}\right)/N^{f}\left(z^{\prime }\right)$ if complete sampling of the forward direction were available. α can be determined in a self-consistent manner. Using Appendix A, we can write the total committor probability in the region $z<z^{\prime }$ and the backward committor probability $z>z^{\prime }$:$$P\left(z\right)=\left\{\begin{array}{cc} \frac{\alpha N^{f}\left(z^{\prime }\right)+N^{b}\left(z\right)-N^{b}\left(z_{min}\right)}{N^{f}\left(z\right)+N^{b}\left(z\right)} & \mathrm{if}\;z<z^{\prime }\\ 1-\frac{N^{b}\left(z_{min}\right)}{N^{b}\left(z\right)} & \mathrm{if}\;z>z^{\prime }. \end{array}\right.$$
If we require that P(z) is continuous at $z=z^{\prime }$, then α = 1 − $N^{b}\left(z_{min}\right)/N^{b}\left(z^{\prime }\right)$. Other ways of determining P(z) for this problem are possible.

### 3.3. The Potential of Mean Force

Typically, the PMFs for ions in channels are calculated in simulations in the absence of electric field using enhanced sampling techniques (see the recent review by Flood, et al. [27]). In contrast, the methods outlined here allow for reconstructing PMF from steady-state simulations with an applied electric field. The underlying PMFs for p7 and GLIC used in the present study were obtained by way of this method (see Appendix A). Since TTX is a bundle of straight α-helices surrounding a featureless water pore, the PMFs for K${}^{+}$ and Cl${}^{-}$ are quite generic, which is characteristic of several very simple channels (see Figure 2a) [24,25]. For K${}^{+}$, the PMF is fairly flat over a wide range of approximately 18 Å inside the channel, which is reminiscent of classical models of ionic conductance in which it is assumed that the PMF is a step function constant inside the channel [1,86]. For Cl${}^{-}$, the PMF is peaked near the center of the bilayer, which can be attributed to the Born barrier experienced by an ion permeating a rigid, featureless non-polar lamella [87]. If an ion is transferred across a membrane through a water-filled pore, the general shape of the PMF remains the same, but the barrier is substantially reduced [87,88]. For TTX, it still remains approximately 1.5 kcal/mol higher than the barrier for K${}^{+}$, which is consistent with a weak selectivity of this channel toward cations.

The PMF for permeation of Cl${}^{-}$ in the OuYang et al. model of p7 [69] is more structured than the PMF for TTX (see Figure 3a). The barriers are low, which explains high chloride current predicted by this model [37]. In contrast to TTX, the barriers to Cl${}^{-}$ permeation in p7 are located near the mouths of the channel due to the presence of positively charged residues at these locations. Compared to permeation of Cl${}^{-}$, the current of K${}^{+}$ in this model is quite low, which indicates that the channel should be anion-selective. Both predicted selectivity and total currents are at variance with electrophysiological data [89], thus contributing to arguments that the proposed high-resolution structure [69] is not native.

The PMF representing permeation of Na${}^{+}$ through GLIC is also markedly more structured than the PMF for ions in TTX (see Figure 2b). The barrier is substantially higher than in the other two channels. As a result, the conductance of this channel is relatively low [70]. This presents a challenge because the number of crossing events in both directions is small. The PMF for Cl${}^{-}$ in this channel is not considered because no crossing events of this ion have been observed in MD simulations.

Taken together, the PMFs considered here are quite different from one another, but are typical of the variety seen in ion channels. In spite of these differences, all three PMFs were successfully reconstructed from non-equilibrium simulations by way of Equations (Equation 12) and (Equation 15) associated, respectively, with the CWDM and CPM methods. The applied voltages were 50 mV for TTX, 140 and −35 mV for p7 and 100 mV for GLIC. For TTX and p7, reconstruction was carried out at the N6, N7 and N8 levels. For GLIC, the number of crossing events at the N6 level was quite small or equal to zero. Thus, only N7 and N8 levels were considered. At the N6 level, 50 and 100 data sets of trajectories were generated for TTX and p7, respectively. At the N7 level, 20 sets of trajectories were generated for TTX and p7, and 50 sets were generated for GLIC. At the N8 level, the number of generated sets was 4, 8 and 25 for TTX, p7 and GLIC, respectively. At each level, all reconstructed PMFs are found to be tightly clustered and their averages at each level are close to the underlying PMF, as shown in Figure 2 and Figure 3a.

In these figures, statistical errors associated with dispersion of the reconstructed PMFs are marked. For $\Delta A(z_{max},z_{min})$, these errors are approximately ±0.3 kcal/mol at the N6 level for TTX and p7 and at the N7 level for GLIC. As expected, they are reduced by approximately a factor of 3 with each level in which the number of sets increases by an order of magnitude. The mean PMFs obtained by way of CWDM and CPM at different levels are quite close to the underlying PMF and the corresponding statistical errors are very similar, indicating that both methods are successful in reproducing the underlying PMFs. Only for GLIC at the N7 level, does the $\Delta A(z_{max},z_{min})$ reconstructed by way of CWDM appear to be systematically underestimated. At this level, no crossing events in one direction are observed for a considerable fraction of data sets, which makes reconstruction of the PMF from Equation (Equation 12) impossible. This systematically biases the sample in favor of sets with higher counts of crossing events and, consequently, lower $\Delta A(z_{max},z_{min})$. From the comparison between the PMFs reconstructed for p7 from trajectories at 140 and −35 mV, it appears that precision of the reconstruction depends somewhat on applied voltage. If the forward and backward densities are well balanced, precision improves.

In CPM, the PMFs can be calculated from one-sided quantities, Equations (Equation 13) and (Equation 14), or by combining them. Here, the latter has been done by way of WHAM, Equation (Equation 15). As can be seen in Figure 3b, this approach yields improved agreement with the underlying PMF. In one-sided formulas, the densities can become quite low near the exit and, as a result, precision in this range suffers.

In summary, both CWDM and CPM provide a reliable means for reconstructing PMFs from non-equilibrium simulations. However, the relation between statistical errors obtained in stochastic and MD simulations is not straightforward. Even if the assumptions of the ED model are satisfied, precision of stochastic simulations is expected to be higher than precision in MD simulations of equivalent length. Specifically, it is usually uncertain if all degrees of freedom perpendicular to the reaction coordinate have been properly equilibrated on the time scale of the simulations. Torsional angles in the side chains of residues lining the pore or motion of whole helices are examples of degrees of freedom that might undergo slow equilibration and, by doing so, influence the calculated PMF and electrophysiological properties. The same concern applies to all other methods for calculating these quantities.

### 3.4. Current-Voltage Dependence

Once the PMFs for the ions permeating the channel have been reconstructed and the committor probabilities for these ions have been calculated for a reference voltage, the full current-voltage (I-V) curves can be calculated from Equations (Equation 19) and (Equation 20) without the need for additional simulations. This is the principal gain in efficiency of the method: the I-V curve can be obtained from a single MD simulation instead of multiple simulations. For example, if constructing the I-V curve required MD simulations at five different voltages in the range of interest the efficiency of our methods would be approximately five-fold. Since numerical results indicate that Equation (Equation 19) yields less accurate results than Equation (Equation 20), this equation will not be further considered. The results for TTX, p7 and GLIC are shown in Figure 4 and Figure 5. The reference applied voltages are the voltages used for reconstructing PMFs, described in the previous subsection. For comparison, currents calculated directly from stochastic trajectories at several voltages are also shown.

As we can see in Figure 4, the agreement between the I-V curves for both K${}^{+}$ and Cl${}^{-}$ in TTX calculated directly and by way of Equations (Equation 18) and (Equation 20) is excellent, even at the N6 level, for the full range of voltages studied here, which extends from −100 to 100 mV. As shown in Figure 6, the I-V curves obtained for different sets of trajectories are closely clustered and deviate from each other only at the largest absolute applied voltages by no more than a few pA.

For p7, the agreement is not as good if the the reference voltage of 140 mV is used for calculating the I-V curve. The Cl${}^{-}$ currents calculated directly and from Equation (Equation 18) agree well for positive voltages, but diverge for negative voltages, away from the reference state. The corresponding statistical errors also increase and become quite large below −50 mV. The source of this disagreement can be traced to the integrand in Equation (Equation 18). As the difference between the reference and the target voltage increases, the exponential term also increases, which magnifies inaccuracies in function f(z). If the reference voltage is chosen to be −35 mV, the agreement over the full range of voltages −150 to 150 mV improves markedly, with modest deviations only at high, positive voltages (see Figure 4). A similar situation was observed for GLIC. For the reference voltage of 100 mV, the I-V curves at the N7 level satisfactorily reproduce currents calculated directly for positive voltages. For negative voltages, the performance of the method progressively deteriorates. Again, if one is interested in an I-V curve that extends to both positive and negative voltages, a different choice of reference voltage may yield significant improvements in accuracy.

As pointed out in the introduction, a number of previous studies have used unrealistically large applied voltages to increase the number of crossing events and, by doing so, improved precision of the calculated currents [15,23,35,37]. Furthermore, as discussed earlier, this may lead to electroporation of the membrane, saturation effects during the intake of ions at the mouth of the channel and involves extrapolation or interpolation to the voltages by way of *ad hoc* procedures of unknown accuracy. The approach developed here is more efficient and accurate and has a substantially stronger theoretical basis than procedures used previously, even though only calculations at the reference applied voltage are necessary. In this approach, the accuracy of the reconstructed I-V curves can be substantially improved through a judicious choice of this reference voltage. This choice depends on the range of voltage that is of interest and on several properties of a channel, in particular its rectification, which characterizes an asymmetry of currents in response to the change in direction of applied voltage. In general, maximizing total ion current through applying high voltage is not the optimal strategy. Instead, it is often better to choose a voltage that yields good statistics in both directions.

### 3.5. Reversal Potential

The reversal potential, $\Delta V_{R}$, is the applied voltage at which there is no net current. If ionic concentrations on both sides of the membrane are equal, $\Delta V_{R}=0$. Experimentally, the reversal potential is measured by maintaining different concentrations on the *cis* and *trans* side of the membrane, and then used in conjunction with the GHK equation to estimate channel selectivity [1,6]. In MD, asymmetric concentrations have to be maintained to measure directly the reversal potential [90], which markedly complicates simulations. We have only considered the situation where the concentrations of ions are the same on both sides of the membrane, as this corresponds to the conditions under which we have carried out MD. We wish to expand this to a range of concentrations.

We expect that the net number of crossing events, from which we calculate the I-V curve, depends on this concentration. If the bulk concentrations are low and the channel is not saturated, then we expect the number of crossing events, and hence the currents, to be linearly dependent on the concentrations of ions. For example, if the concentration is doubled on both sides of the membrane, the net currents will also double. Under these assumptions, we can calculate the reversal potential from our formalism. We simply need to scale the fluxes of all ion types on one side of the membrane to match the desired concentration difference.

The K${}^{+}$/Cl${}^{-}$ selectivity of TTX obtained from the MD simulations is 2.2 [24,33]. Using currents scaled by 5:1 in the I-V reconstruction from Equation (Equation 20), we obtain a reversal potential of −9 mV. This corresponds to a GHK selectivity of 1.7, which is reasonably close to the selectivity found in MD. Note that experimentally, the reversal potential is −27 mV, corresponding to a K${}^{+}$/Cl${}^{-}$ selectivity of 6 estimated from the GHK Equation [6]. This cannot be compared directly to our results because the actual channel structure is unknown, there are uncertainties due to force fields, and the GHK equation itself is an approximation.

## 4. Conclusions

Stochastic simulations were used to investigate the reliability of two new methods to calculate PMFs for ion transport across transmembrane ion channels and electrophysiological properties of these channels within the general framework of the electrodiffusion model. Both methods have the desirable features that only simulations at a single voltage are needed and information on the diffusivity is not required. In CPM, knowledge of the committor probability is required. Stochastic simulations containing 10${}^{6}$ trajectories were shown to have similar numbers of crossing events for models of TTX and p7 in MD simulations of 1–2 μs in length. Analysis of 50 or 100 of such simulations indicate that errors in the free energy profiles are approximately ±0.3 kcal/mol. For a model of a slow channel, GLIC, 10${}^{7}$ trajectories, which approximately corresponds to MD simulations of 10 μs in length, are needed to achieve a similar statistical error. For both TTX and for p7 at lower applied voltages, CPM and CWDM yield similar results. In CWDM, one-sided fluxes are used directly, and for cases in which few crossing events are observed in one direction, either due to large applied voltages, such as p7 at ±140 mV, or because the channel is rectifying, such as GLIC, CPM performs better because two-sided quantities are employed in this method. Similarly, even though one-sided CPM calculations are possible, the errors near the end of the channel become substantial because the density becomes quite small, yielding large relative errors.

Stochastic simulations were also used to investigate the reliability of a new expression to calculate the ionic currents at different voltages, ΔV, given knowledge of the PMF, committor probabilities and density profiles at a reference voltage $\Delta V_{\mathrm{ref}}$. We found that the I-V dependence could be reconstructed over a range of ±100 mV, with respect to the reference voltage. Judicious choice of $\Delta V_{\mathrm{ref}}$ can markedly improve the accuracy of the reconstruction. Specifically, the I-V reconstruction for p7 is much better for $\Delta V_{\mathrm{ref}}$ = −35 mV than for $\Delta V_{\mathrm{ref}}$ = 140 mV. Although much of the error can be attributed to the large voltage ramp for voltages away from $\Delta V_{\mathrm{ref}}$ (3.2 kcal/mol at 140 mV), some of the error is due to the poor statistics in the direction against the field. This is also evident in the reconstruction of the I-V curve for GLIC, for which some simulations yielded no crossing events against the field.

Common goals of simulations of ion channels are to obtain the free energy profiles of ions translocating the channel and to determine electrophysiological properties of the channel. In some instances, a reliable estimate of the numbers of crossing events, from which the ionic currents can be calculated, is difficult to obtain from MD even for long simulation times. We have shown that the new methods perform very well both to obtain reliably the free energy profile across the channel and to allow for accurate determination of the I-V curves. In the latter case, it is desirable to use a reference voltage that yields good crossing statistics in both directions rather than a voltage that maximizes the total number of crossing events. In summary, if transport of ions through a channel can be satisfactorily described by the ED model, the new methods offer substantial reductions of computational effort without sacrificing accuracy. Our approach is amenable to extensions in which the advantages of MD and stochastic simulations are further combined on reliable theoretical grounds.

## Figures and Tables

**Figure 1 entropy-23-00571-f001:**
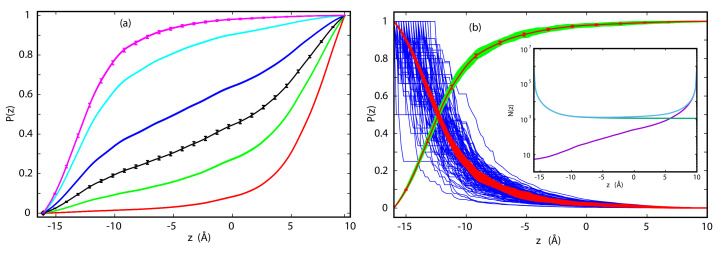
(**a**) Committor probabilities for Cl${}^{-}$ in p7 at −140 mV (red), −70 mV (green), −35 mV (black), 0 V (blue), 70 mV (cyan) and 140 mV (magenta). Error bars are shown for the N6 data sets at −35 mV and 140 mV; (**b**) Committor probabilities for p7 at 140 mV from the N6 data set for 1-sided forward (green) and backward (blue) trajectories, respectively, 2-sided data set in the backward direction (red lines), and average in the forward direction with error bars (red symbols). In the inset we show the number of first passage trajectories to reach *z* for one N6 data set in the forward (green) and backward (magenta) directions and the total (light blue).

**Figure 2 entropy-23-00571-f002:**
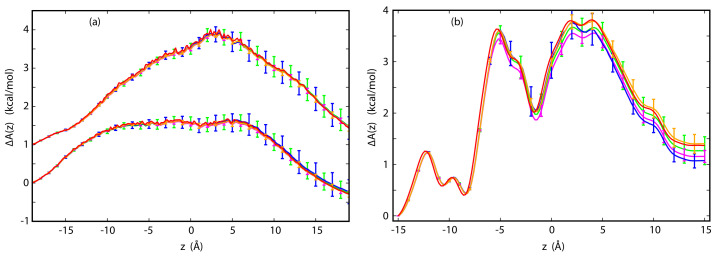
(**a**) PMFs for K${}^{+}$ (lower curves) and Cl${}^{-}$ (upper curves) in TTX from stochastic simulations with an applied voltage of 50 mV. The PMFs have been reconstructed by way of CWDM at the N6 (blue) and N7 (gold) levels or by way of CPM at the N6 (green) and N7 (magenta) levels; (**b**) PMF for Na${}^{+}$ in GLIC from stochastic simulations with applied voltage of 100 mV. The PMF has been reconstructed by way of CWDM at the N7 (blue) and N8 (gold) level or by way of CPM at the N7 (green) and N8 (magenta) level. In both panels, the underlying PMF is in red.

**Figure 3 entropy-23-00571-f003:**
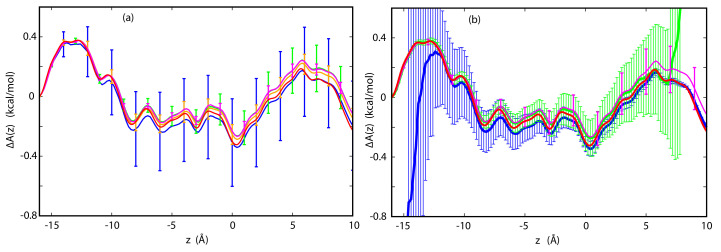
(**a**) PMF for Cl${}^{-}$ in p7 from stochastic simulations with an applied voltage of 140 mV. The PMFs have been reconstructed by way of CWDM at the N6 (blue) and N7 (gold) levels or by way of CPM at the N6 (green) and N7 (magenta) levels. The input PMF (red) is shown for reference. PMFs at the N8 level are not shown, as they coincide with the underlying PMFs and statistical errors associated with this level arequite small and are poorly visible at this scale; (**b**) PMFs for P7 reconstructed by way of one-sided forward trajectories (green) using Equation (Equation 13) and backward trajectories (blue) using Equation (Equation 14) from stochastic simulations at the N6 level with applied voltage of 140 mV. Two-sided reconstruction (magenta) and the underlying PMF (red) are shown for comparison. Note that one-sided, but not two-sided reconstructions are burdened with large errors at the ends.

**Figure 4 entropy-23-00571-f004:**
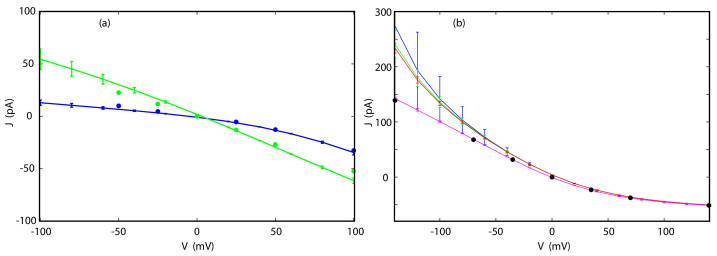
(**a**) I-V curves for K${}^{+}$ (green) and Cl${}^{-}$ (blue) in TTX reconstructed from simulations at 50 mV at the N6 level. Blue and green dots are currents obtained from direct simulations at specific voltages.; (**b**) I-V curves for Cl${}^{-}$ in p7 reconstructed from simulations at 140 mV at the N6 (blue), N7 (green) and N8 (red) level, and for −35 mV at the N6 level (magenta). N7 and N8 curves are not shown because they are almost identical to the N6 results. Black dots are currents obtained from direct simulations at specific voltages. All reconstructions were done using the PMFs obtained by way of CPM. The results of reconstructions using the PMFs from CWDM are not displayed because they are nearly identical.

**Figure 5 entropy-23-00571-f005:**
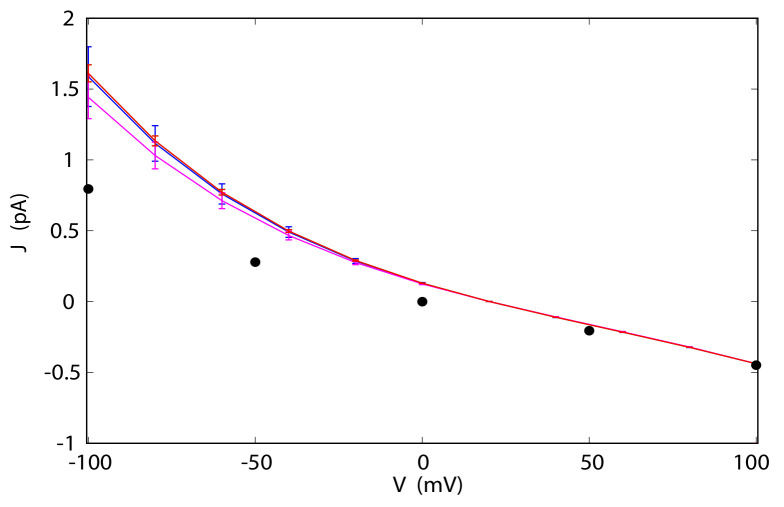
I-V curves for Na${}^{+}$ in GLIC reconstructed from simulations at 100 mV at the N7 level with PMF from CPM (blue), at the N7 level with PMF from CWDM (magenta), and N8 with PMF from CWDM (red). N8 with CPM (not shown) is almost identical to N7 CPM. Black dots are currents obtained from direct simulations at specific voltages.

**Figure 6 entropy-23-00571-f006:**
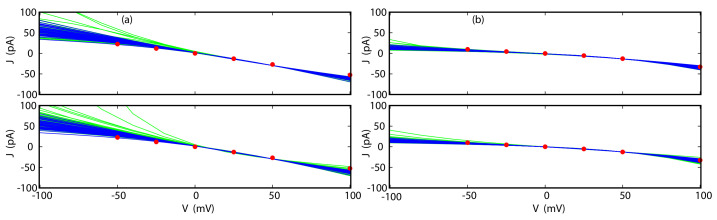
Reconstructions of I-V curves in TTX from individual sets of trajectories for K${}^{+}$ (**a**) and Cl${}^{-}$ (**b**). The PMFs were obtained from CPM (upper panels) or CWDM (lower panels). The curves were calculated by way of Equation (Equation 20) (blue) or Equation (Equation 18) (green). All reconstructions were carried out from simulations at applied voltage of 50 mV at the N6 level. Note that blue curves, but not green curves, are tightly clustered together indicating that Equation (Equation 20) is more accurate than Equation (Equation 18).

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
