# Peer review of "Electrophysiological Properties from Computations at a Single Voltage: Testing Theory with Stochastic Simulations"

_entropy, 2021, doi:10.3390/e23050571_

Round 1

Reviewer 1 Report

The manuscript by Pohorille and Wilson addresses a classical problem from Biophysics, but employing an interesting and quite novel approach that combines together classical macroscopic equations of (steady) Fokker-Planck type with microscopic Molecular Dynamics Simulations (a widely used, and abused, tecnique among computational Biopyisicists). This quite innovative approach (early attempts have been honestly cited by the authors) looks promising and it is scientifically sound. I have, however, a couple of comments aimed at improving the quality of the work.

A) I'm afraid that inexperienced readers may wonder about the missing of a coupled Poisson's equation to the transport equation (1) as generally done in macroscopic analytical model. The Poisson's equation enables to calculate self-consistently the density of the i-th ion ρi(z) under the effect of all the charged species in the system. In the present model the density is calculated by a MD technique, an approach that consider both electrostatic (through the Ewald's Sum) and non-electrostatic interactions. Albeit this is an obvious observation, I believe that this point deserves a short comment.

B) I also suggest to explain why ρi(z) is independent of the local diffusion coefficient D(z) (at least, in steady conditions). 

C) The authors claim that they made "single channel" simulations. Since they impose "periodic boundary conditions", strictly speaking they are simulating a 2D infinite array of channels embedded in a lipid membrane. Only if the ionic current does not appreciably change by varying the size of the unit cell (i.e., by increasing the number of lipid and water molecules), this claim is correct. A simple and easy control experiment would solve the question.

D) Are the MD results sensitive to the employied force-field? 

E) The quality of the paper could dramatically improve if the authors widen their analysis to others pore-forming peptides.

Round 2

Reviewer 1 Report

The revised manuscript by Wilson and Pohorile contains significant improvements. For this reason, I believe it could be published in Entropy in the present form. However, I encourage the authors to revise the Introducion and conclusion  sections  in order to better clarify the relevance of their work in  the context of a very over-crowded field. Asa corollary, a more accurate description of the existing wide literature is warranted.

Round 3

Reviewer 1 Report

The revised manuscript by Wilson and Pohorile meets most of my previous concerns. I believe it can be published in the present revised version.